# Electrophoresis-Aided Biomimetic Mineralization System Using Graphene Oxide for Regeneration of Hydroxyapatite on Dentin

**DOI:** 10.3390/ma15010199

**Published:** 2021-12-28

**Authors:** Ingrid Patricia Khosalim, Yu Yuan Zhang, Cynthia Kar Yung Yiu, Hai Ming Wong

**Affiliations:** Paediatric Dentistry and Orthodontics, Faculty of Dentistry, The University of Hong Kong, Hong Kong SAR, China; ingridpk@connect.hku.hk (I.P.K.); zyuyuan@hku.hk (Y.Y.Z.); ckyyiu@hku.hk (C.K.Y.Y.)

**Keywords:** graphene oxide, hydroxyapatite, electrophoresis

## Abstract

Graphene oxide (GO) is an emerging luminescent carbon nanomaterial with the ability to foster hydroxyapatite (HA). A specially designed electrophoresis system can be used to accelerate the mineralization process. The aim of this study was to promote HA crystal growth on demineralized dentin using a GO incorporated electrophoresis system. GO was successfully synthesized by carbonization of citric acid and its presence was confirmed by Fourier transform infrared and UV-visible spectrophotometry evaluation. Dentin slices were placed in demineralized solution and divided into control (without the electrophoresis system) and experimental group. Demineralized dentin slices in the experimental group were remineralized using the electrophoresis system for 8 h/1.0 mA, with one subgroup treated without GO and the other with GO. Energy dispersive spectroscopy evaluation showed that the calcium/phosphate ratio of the crystal formed in control and experimental group with addition of GO was close to natural hydroxyapatite. However, scanning electron microscopy evaluation showed that the exposed dentinal tubules were occluded with rod-like crystals, which is similar to native enamel morphology, in the experimental group with addition of GO compared to the flake-like crystal in the control group. Mechanical evaluation revealed that the nanohardness and modulus of remineralized dentin were significantly higher in the experimental group. In conclusion, GO is a promising material to remineralize dentin and the introduction of an electrophoresis system can accelerate its process.

## 1. Introduction

Tooth structure consists of enamel and dentin-pulp complex. Enamel acts as a barrier that protects dentin-pulp complex from physical, thermal, and chemical stimuli [1]. Dentin is vital tissue consisting of dentinal tubules, extension of odontoblasts, and formation of dentin-pulp complex. It has the capacity to respond to physiologic and pathologic stimuli [2].

Hydroxyapatite (HA) is an inorganic material found in 70% of dentin, along with other trace elements. HA exhibits bioactive properties because it has natural response of cells and mineralization process [3]. HA-particles have high polarity and the ability to bind to both collagen and HA from dentin. HA-particles occlude dentin tubules by being pressed into the tubule opening. HA-particles that bind to the tubules will be mineralized by calcium and phosphate ions from saliva [4,5].

Biomimetic mineralization is an innovative way to mimic natural process of mineralization. It aims to introduce calcium and phosphate ions on demineralized lesion which results in remineralization [6]. A number of biomimetic mineralization methods have been reported, such as a hydrothermal method [1], application of acidic calcium phosphate paste containing hydrogen peroxide and phosphoric acid [7]. Other methods include the use of bioactive glass [8], surfactants [9], nano-HA and proline [10], amelogenin [11,12], gelatin [13], dendrimer [14], ethylenediaminetetraacetic acid (EDTA) [15], polyethylene oxide, polyacrylamide [16], and polydopamine [17]. The use of induced pluripotent stem cells (iPSCSs) has been explored and proposed as an alternative in regeneration of mineralized tooth components or supporting tissue [18]. In vitro study showed that biomimetic mineralization was able to regenerate HA [19]. However, in the aforementioned methods, the rate of remineralization is very slow, limiting the clinical application.

Electrophoresis-aided mineralization system is one of the many biomimetic mineralization methods for regenerating HA [2]. The electrophoresis system transports ions better through gels or solutions compared to only diffusion. Agarose hydrogels can be used to promote the synthesis of HA. [20,21]. The electrophoresis system allows ions to move in a one-dimension direction specifically. Advanced movement of calcium and phosphate ions in the agarose hydrogels was observed after electric current was generated. A novel electrophoresis-aided mineralizing system was created to promote the diffusion of calcium and phosphate ions and to accelerate the speed of mineralization [2]. Numerous in vitro studies have proved the efficacy of the electrophoresis system. A study by Wu et al. successfully remineralized a completely demineralized dentin collagen matrix by using the electrophoresis-aided biomineralization system. Another study by Zhang et al. conducted in vitro and in vivo evaluation in which it was observed that electrophoresis could improve the remineralization process of casein phosphopeptide-amorphous calcium phosphate (CPP-ACP) and the demineralized enamel surface was completely remineralized after treatment of 5 h [20,22,23].

Graphene oxide (GO) is an emerging luminescent carbon nanomaterial. It is an atomically thin sheet of graphite and oxygen-containing functional groups are covalently decorated, either on the basal plane or at the edges. GO has favorable characteristics such as biocompatibility, low toxicity, strong mechanical strength, and high elasticity and flexibility [24]. In addition, GO has the ability to foster hydroxyapatite, which is an important component in mineralization process. GO could promote remineralization with express HA growth, forming a coating of homogenous and compacted HA/GO and consisting flake-shaped HA crystals with marked elevated calcium-phosphate ration [7,25,26,27].

With the ability of GO to foster HA growth and electrophoresis system to accelerate HA formation, the aim of this research was to promote HA crystal growth using GO incorporated electrophoresis system. We hypothesized that HA crystal growth could be promoted by the use of GO incorporated electrophoresis system.

## 2. Materials and Methods

### 2.1. Specimen Preparation

This study was approved by The University of Hong Kong/Hospital Authority Hong Kong West Cluster Institutional Review Board (IRB UW17-009). This study was carried out in accordance with approved guideline for research involving human subjects. Extracted third molars with sound enamel were obtained from participants with patients’ written informed consent. Teeth were rinsed with normal saline, cleaned ultrasonically to remove the debris, and fixed in 10% buffered formalin (Sigma-Aldrich, St. Louis, MO, USA) for at least 3 months. Dentin slices of 2 mm thickness were prepared perpendicular to the longitudinal axis of the tooth using a low-speed diamond saw (IsoMet Low Speed Saw, Buehler, Lake Bluff, IL, USA). Silicon carbide papers with 600-, 1200-, 2400-, and 3000-grit were used for polishing. The slices were ultrasonically cleaned and stored at 4 °C in deionized water. 

### 2.2. Lesion Formation and pH Cycling

Two dentin slices were used to study sound dentin. Dentin slices were placed in a demineralization solution (2.2 mM CaCl_2_ · 2H_2_O, 2.2 mM KH_2_PO_4_, 50 mM acetate, pH 4.4) for 96 h at 37 °C to create lesions of 70–100 mm deep. After the demineralization process, the dentin slices were rinsed thoroughly with deionized water. Dentin slices were pH cycled at room temperature through a 30 min immersion in demineralization solution (1.5 mM CaCl_2_, 0.9 mM KH_2_PO_4_, 50 mM acetate) at pH 5.0 followed by a 10 min immersion in remineralization solution (20 mM 4-(2 hydroxyethyl)-1 piperazineethanesulfonic acid (HEPES), 1.5 mM CaCl_2_, 0.9 mM KH_2_PO_4_, 150 mM KCl) at pH 7.0. The pH-cycling procedure was performed six cycles per day for 8 days [28]. All of the solutions were freshly prepared for use in the experiment. Dentin slices were stored in deionized water at 4 °C overnight. At the end of the cycling period, all specimens were washed with deionized water, and air dried.

### 2.3. Synthesis of GO

Pyrolyzing citric acid was used to synthesize GO. First, 1 gr of citric acid (Sigma-Aldrich, St. Louis, MO, USA) was put into a 5 mL beaker and heated to 200 °C. After 5 min, the citric acid was liquefied and the color changed from colorless to yellow. After 30 min, the color changed to orange. The heating was kept until it changed into black liquid in 100 min, suggesting the formation of GO. The black liquid was then added to KH_2_PO_4_ solution (Sigma-Aldrich, St. Louis, MO, USA) and by 80 mg/mL KOH solution to obtain 1.55 M KH_2_PO_4_ solution 0.8 wt% GO with pH 6.5. [29,30]. The solution was stored at 4 °C before use.

### 2.4. Characterization of GO

The Fourier transform infrared (FTIR) spectra were obtained on a FTIR spectrophotometer (Thermo Nicolet 360, GMI, Ramsey, NJ, USA) equipped with an attenuated total reflection (ATR) sampling accessory. Spectra were recorded at room temperature, in the wavenumber range of 4000–500 cm^−1^, with an incident laser power of 1 mW and a minimum resolution of 4 cm^−1^. FTIR were used to identify pure substances, mixtures, impurities, and compositions of GO. UV–Vis absorption was characterized by a UV/Vis/NIR spectrophotometer (Lambda 750, PerkinElmer Inc., Akron, OH, USA).

### 2.5. Preparation of the Mineralizing Medium in Agarose Hydrogel

A CaCl_2_-agarose hydrogel was prepared by mixing 1.0 g agarose powder (Regular Agarose G-10, BIOWEST, Nuaille, France) into 100 mL of a 0.13 M CaCL_2_ solution (CaCl_2_·2H_2_O, Sigma-Aldrich, St. Louis, MO, USA). A KH_2_PO_4_-agarose hydrogel was prepared by mixing 1.0 g agarose powder into 100 mL of a 1.55 M KH_2_PO_4_ solution (Sigma-Aldrich, St. Louis, MO, USA) containing 500 ppm fluoride (Sigma-Aldrich, St. Louis, MO, USA). A GO-agarose hydrogel was prepared by mixing 1.0 g agarose powder into 100 mL of 1.55 M KH_2_PO_4_ solution containing 0.8 wt% GO (Sigma-Aldrich, St. Louis, MO, USA) and 500 ppm fluoride (Sigma-Aldrich, St. Louis, MO, USA). The pH value of the solutions was adjusted to 6.5 using 0.1 M KOH and 0.1 M HCl. The mixtures were enhanced for 30 min and then heated to 100 °C until agarose was completely dissolved. 

### 2.6. Regeneration of Hydroxyapatite in Agarose Hydrogel Aided by Electrophoresis

Six dentin slices were used to study the structure and mechanical properties of completely demineralized dentin. Eighteen dentin slices were allocated into experimental (*n* = 12) and control group (*n* = 6). Dentin slices in experimental group were equally allocated into group A (without the addition of GO) and group B (GO with 0.8%). In group A and control group, dentin slices were placed between the CaCl_2_ agarose hydrogel and the KH_2_PO_4_ agarose hydrogel [23]. In the group B, dentin slices were placed between the CaCl_2_ agarose hydrogel and the GO-phosphate agarose hydrogel which were put into the two sides of the tube. The tubes were then connected to the plastic cells. Electrodes were set into the bottom of the cells, which were filled with 0.9% NaCl solution to enhance the electrical conductivity. Dentin slices in group A and B were treated with electrophoresis system. The electric current was maintained constant at 1.0 mA, during electrophoresis [2]. The gels and NaCl solution were refreshed every 2 h, and their exchange defined the completion of a cycle of mineralization. The dentin slices were cleaned ultrasonically for 2 min after each cycle. The sample was taken out after two, four, and six cycles for assessment and then stored at 4 °C for further characterization.

### 2.7. Characterization and Evaluation of New Crystal

The morphology of new crystals and the chemical analysis with respect to the Ca/P ratio of the remineralized dentin slices were evaluated using field-emission scanning electron microscopy (FE-SEM) and energy dispersive spectroscopy (EDS) (Hitachi S4800, Hitachi Ltd., Tokyo, Japan), respectively. Before the SEM, and EDS evaluation, all samples were dehydrated with gradual ethanol and dried in critical evaporator. The mechanical properties of remineralized dentin slices was evaluated by nanoindentation test. The elastic modulus and nanohardness of the dentin slices were evaluated by nanoindentation measurement (TI900 Nanomechanical Test Instrument, Hysitron Inc., Minneapolis, MN, USA). The duration for loading and unloading were both 15 s, with holding time 10 s under the maximum applied force of 40 mN. For each of the samples, six sites were tested. The elastic modulus and nanohardness were calculated from the force-displacement curves by using Triboscan Quasi software. All data obtained were recorded and analyzed with statistical software (SPSS Statistic 25; IBM, Armonk, OH, USA). Differences were considered significant at *p* < 0.05 (*t*-test). Data were expressed as mean ± standard deviation.

## 3. Results

### 3.1. The Synthesis of GO

In this study, the band at 3390 cm^−1^ was attributed to the OH stretching vibration due to the presence of surface hydroxyl groups (Figure 1). The band at 1715.31 cm^−1^ was the characteristic of the C=O stretch of the carboxylic acid group in citric acid (Figure 1, line a). The C-O bonds were attributed by 1290.50 cm^−1^, presenting the oxide functional groups in the synthesized GO and confirming the successful carbonization of citric acid into GO (Figure 1, line b).

The UV-Vis region of energy for electromagnetic spectrum was observed in the range of 200–350 for UV and 350–700 for Vis. The absorption in the UV-Vis region is related to electric transitions in particles. Electronic transition occurs from bonding and/or non-bonding orbitals to the antibonding orbital when electromagnetic radiation absorbed. After carbonization, the color of citric acid changed into visible dark, and its UV-Vis absorption spectrum was also changed, due to the difference in molecular structure between citric acid and GO (Figure 2).

### 3.2. Characterization of Remineralized Dentin Dlices

Figure 3 illustrates the remineralized dentin slices after 8 h of remineralization. In the control group without the application of electrophoresis-aided system, only sporadic crystal was observed on the dentin surface. The vast majority of the dentinal tubules remained open (Figure 3a). In the experimental group with the aid of electrophoresis-aided system and addition of GO, acid-etched dentin slices were fully remineralized, and their surfaces were covered by the newly formed crystal layer (Figure 3b). The EDS spectrum showed the presence of higher amounts of carbon and oxygen (Figure 3d). This confirmed that the membrane contained GO to its surface, compared to pure HA (Figure 3c). Other elements are found as impurities on the membrane surface. Table 1 showed the concentration (wt.%) of the major mineral component, suggesting higher concentration (wt.%) of carbon and oxygen in the experimental group with addition of GO compared to pure control group.

After introduction of GO, crystals with different morphology were observed. The flake-like crystals were formed on the surface of acid-etched dentin in group A without the addition of GO (Figure 4), while rod-like crystals were obtained after the addition of GO (Figure 5).

### 3.3. Mechanical Evaluation

After 8 h of remineralization, the nanohardness of remineralized dentin slices in group A was significantly higher than that of acid-etched dentin slices (0.347 ± 0.128; *p* < 0.0001) and reached to the level of nanohardness of native dentin slices (1.068 ± 0.042; *p* = 0.885) (Figure 6a). The modulus of remineralized dentin slices in group A (25.27 ± 1.621) was significantly higher than that of acid-etched dentin slices (7.275 ± 0.213; *p* < 0.0001) and that of native dentin slices (20.60 ± 1.146; *p* = 0.04) (Figure 6b).

The nanohardness of remineralized dentin slices in group B (1.358 ± 0.094) was significantly higher than those in group A (*p* = 0.0184), while the modulus of remineralized dentin slices in group B (27.48 ± 1.319) was similar to those in group A (25.27 ± 1.621; *p* = 0.316).

## 4. Discussion

Fourier transform infrared spectroscopy (FTIR) utilizes infrared radiation to detect functional groups found in materials. FTIR measures the absorption of infrared produced by the covalent bond in each molecule. Each bond and functional group absorb different frequencies, resulting in different transmittance patter for each molecule. The spectrum is recorded on *X* and *Y*-axis as wavelength (cm^−1^) and % transmittance, respectively [31]. In this study, the result is in accordance with the study by Yang et al. and Hui et al. which showed the carbonization of citric acid into GO [32,33]. It was observed that no characteristic absorption bands of aromatic compound (stretching vibration of C–H in aromatic rings around 3000–3100 cm^−1^, skeletal vibration of aromatic rings around 1450–1650 cm^−1^) was found in the FTIR spectrum. This suggested that there was no aromatic compound found in the GO synthesized (Figure 1).

The graphene synthesis process is a graphene fabrication process based on the desired size and quality. There are various approaches to synthesize GO and its derivatives. Methods such as mechanical cleavage (exfoliation), chemical synthesis, chemical exfoliation, epitaxial growth, and thermal chemical vapor deposition are widely known. Some other methods, namely electrochemical peeling, microwave synthesis, and carbon nanotube unzipping, have been reported. When massive production of graphene is required, methods such as top-down and bottom-up process of single layer graphene, bilayer graphene, and a few layers of graphene can be used. In top-down process, exfoliation or separation of highly ordered pyrolytic graphite or its derivatives was used to synthesize graphene sheets. On the other hand, graphene sheets were synthesized by building up nanoscale material via atomic or molecular arrangement of carbon in bottom-up process. [34]. In the present study, the GO was synthesized using bottom-up process by regulating carbonization degree of citric acid.

Different to that of citric acid, the unique atomic and electronic structure of variable sp^2^/sp^3^ fractions exists in GO [35]. GO can emit near-infrared, visible and ultraviolet photoluminescent, behaving in the form of the luminescence centers or chromophores, through the existence of geminate recombination of e-h pairs in the sp^2^ clusters [36,37].

GO solution can release blue light (460 nm) upon stimulation of 365 nm UV beam, which corresponds with the results in this study [37]. The UV-Vis adsorption of synthesized GO showed its maximum emission wavelength was 365 nm; meanwhile, a shoulder around 320 nm was observed, which could be due to the n–π* transitions of C=O [38]. This is consistent with the study conducted by Marcano et al. who examined improved synthesis of GO which showed that the UV- vis spectrum has a similar shoulder around 300 nm due to n–π* transitions of the carbonyl groups [39].

Another study by Khalili et al. observed the UV-Vis spectrum of GO, where the main spectrum of it has strong absorption point at 233 nm, which is associated with the π–π* transition of the CC conjugated aromatic domain and weak absorption (shoulder) at 305 nm. The UV-vis spectrum provides evidence of the presence of a number of oxygen functions, such as hydroxyl, carboxyl, epoxide, and carbonyl in grapheneoxidants [40].

Energy dispersive spectroscopy aims to determine the surface elemental composition of GO flakes. This method provides valuable information about the elemental composition of GO flakes. Since GO was acquired through carbonizing citric acid, thus it was observed that the carbon peak of GO-HA was stronger than that of pure HA.

Based on our previous study, the native dentin slices could be completely demineralized after treatment with 30% phosphate acid for 15 s. All the dentin tubules and dentin collagen fibers could be exposed [2]. Due to the existence of extrinsic electric current, the mineralization speed was accelerated. The electric field is a method for manipulating bioparticles with strong control and ability with high efficiency, one of which is electrophoresis. Electrophoresis can deliver ions within teeth by forming electrical fields in the direction of low amperage electric currents. This is in accordance with the study by Zhang et al. which showed that electrophoresis introduced in their study in combination with CPP-ACP can accelerate the rate of remineralization [23]. A study by Wu et al. demonstrated an XRD pattern from crystals growing on the surface of dentin that are completely demineralized after remineralization for six cycles in a biomimetic mineralization system with the help of an electric field [22].

Compared to the flake-like crystals, rod-like crystals have a similar morphology to the crystals in native enamel. GO nanosheet is a monolayer of carbon atoms with condensed honeycomb structures, containing numerous reactive oxygen functional groups. Due to its excellent functionalization, GO has been widely applied as the precursor for biomimetic synthesis, especially for HA-based composite materials. Therefore, we could observe different crystal morphology being obtained after the addition of GO.

In the study by Nizami et al., GO-coated dentin slices were remineralized which resulted in almost completely sealed dentinal tubules compared to untreated dentin slices showing more open dentinal tubules (openings approximately 2 μm in diameter). The results showed that the dentinal tubules were completely sealed with GO-Ag-CaF_2_, almost completely sealed with GO alone, GO-Ag, GO-CaF_2_, GO-Ca_3_(PO_4_)_2_, and partially sealed with GO-Zn. In our study, only GO was used to enhance the remineralization, but the electrophoresis system was used to accelerate the process. Our finding was in line with the study by Nizami et al., in which the remineralization was observed for the group with the addition of GO. However, 8 h of mineralization time was needed in our study compared to 24 h of mineralization time in the study by Nizami et al. [41].

Graphene oxide has been known due to its excellent mechanical properties, which is attributed to the hexagonal lattice formed by stable sp^2^. The atoms in its honeycomb-like structure are combined with the sp^2^ clusters [42,43]. Graphite is a material consisting of three-dimensional carbon that has millions of layers of graphene. Graphite oxidation uses a strong oxidizing agent and its function is to expand the separation of the layers and also create a hydrophilic material. GO is a single-atom layered material composed of carbon, hydrogen and oxygen molecules through the oxidation of inexpensive and abundant graphite crystals [30,44,45]. Furthermore, GO has good biological properties and has been widely used in biomedical field, including drug delivery [46], cancer therapy [47,48], biosensing [49], tissue engineering [50], and bioimaging [51]. The use of GO can enhance cell proliferation and differentiation characteristics. A study by Chen et al. showed that the iPSCs cultured on GO surfaces spontaneously differentiated into ectodermal and mesodermal lineages uneventfully. The study concluded that GO is a promising material that can be used for iPSCs culture [52]. Evidence demonstrated that the mechanical performance and biocompatibility of HA had been improved significantly by reinforcement with GO [53].

In this study, it was observed that the result of experimental group B (with addition of GO) showed better result, suggesting that the addition of GO could effectively improve the nanohardness of remineralized dentin slice. Compared to enamel, dentin is more difficult to remineralize due to its different structure. Enamel has a high concentration of inorganic HA. The apatite crystals of demineralized enamel will continue to attract calcium and phosphorous ions which can facilitate remineralization. On the contrary, dentin consists of more organic substances including collagen fibrils. After demineralization, only a small amount of dentin can be reversed by surface remineralization, as well as growth and reproduction of crystalline cells. The mineralization of dentin occurs through non-collagenous protein regulation, deposition of calcium and phosphorus ions on dentinal collagen fibrils, and lastly formation of HA [54]. This difference is also related to the pattern of remineralization associated with the lower number of remaining crystals and the higher surface area of organic matrix (mainly type I collagen) found on the demineralized dentin surface. Furthermore, some studies showed that a portion of non-collagenous soluble protein released from dentin can inhibit remineralization [55].

The result of this study proved the ability of GO to promote HA crystal growth and the aid of electrophoresis system could accelerate the process. To date, there has been no clinical study conducted and it can be explored in future studies. Considering the shortened duration required to remineralize the tooth surface and the safety, electrophoresis has the potential to be applied clinically.

## 5. Conclusions

The introduction of electrophoresis system could significantly improve the generation of HA by GO. The occlusion of dentinal tubules with enamel-like tissue was observed on the dentin surfaces treated with electrophoresis with addition of GO. This study presented a promising way to mineralize dentin biomimetically.

## Figures and Tables

**Figure 1 materials-15-00199-f001:**
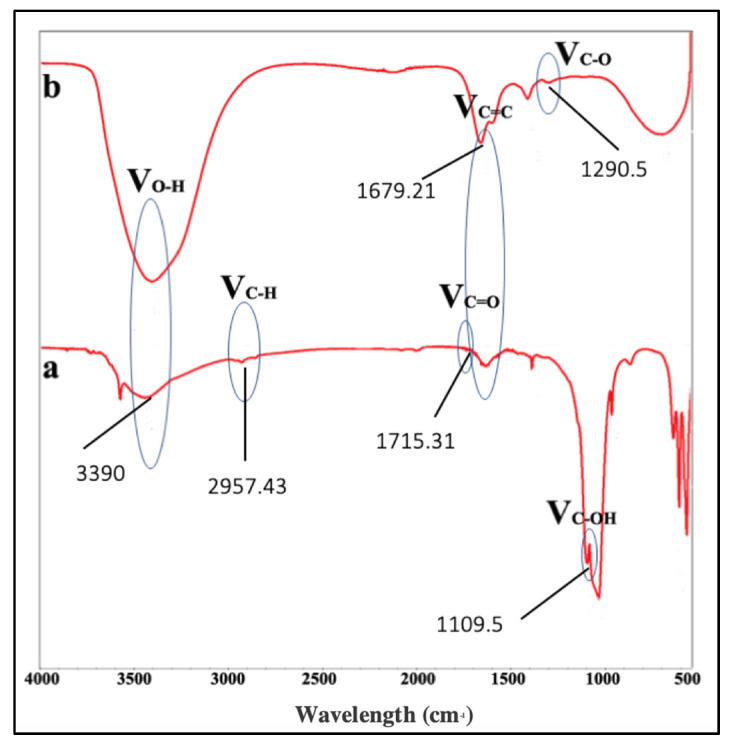
FTIR evaluation of citric acid and graphene oxide (GO). Line (**a**) shows the characteristic of the C=O stretch of the carboxylic acid group in citric acid. Line (**b**) presents the oxide functional groups in the synthesized GO and confirms the successful carbonization of citric acid into graphite oxide.

**Figure 2 materials-15-00199-f002:**
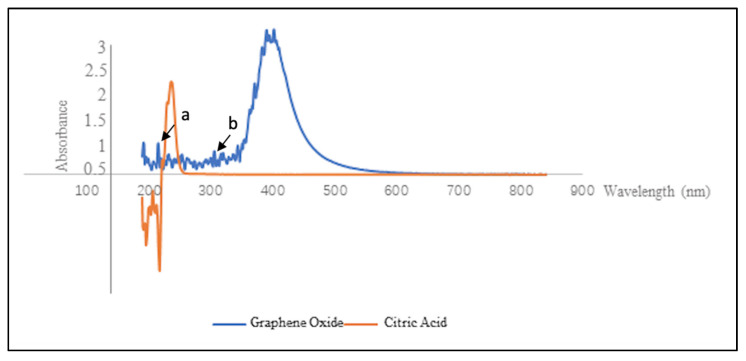
UV–visible adsorption spectrum of citric acid and GO. Arrow (**a**) represents π–π* transitions of aromatic C-C bonds. Arrow (**b**) represents n–π* transitions of C=O.

**Figure 3 materials-15-00199-f003:**
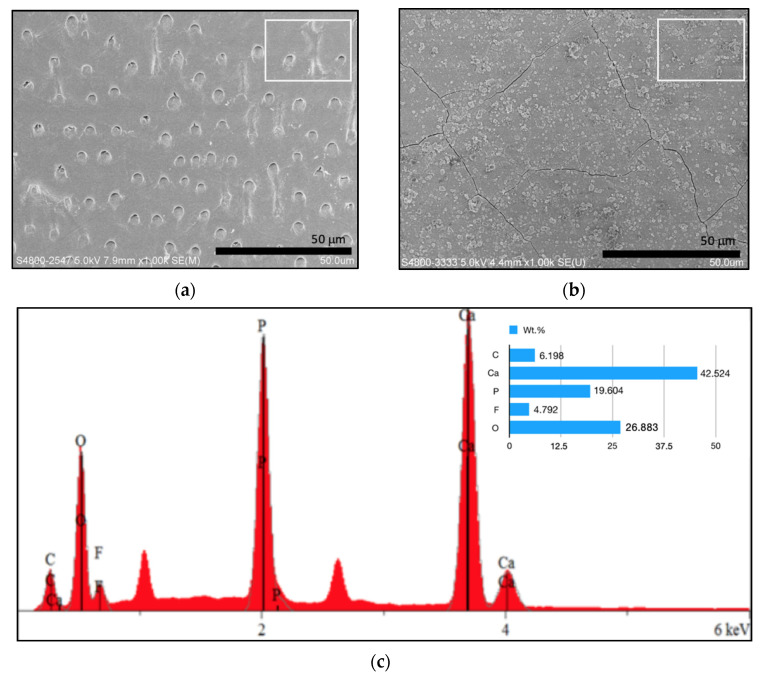
SEM micrograph of remineralized dentin slices; (**a**) dentin surface after remineralization without the aid of electrophoresis-aided system; (**b**) dentin surface after remineralization treated with electrophoresis-aided system and addition of GO; (**c**) energy dispersive spectroscopy (EDS) of remineralized area (rectangle) in (**a**); (**d**) EDS of remineralized area (rectangle) in (**b**).

**Figure 4 materials-15-00199-f004:**
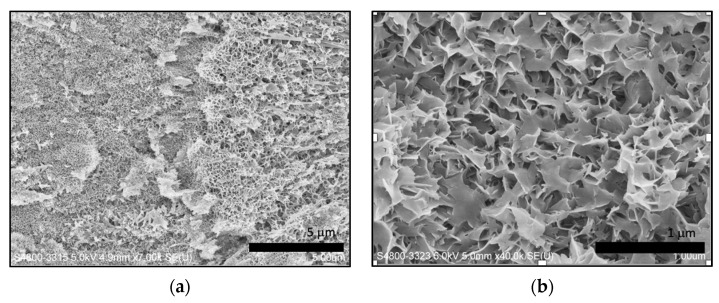
SEM micrograph of remineralized dentin surface in the experimental group without addition of GO: (**a**) with 7.00 k × magnification; (**b**) with 40.0 k × magnification.

**Figure 5 materials-15-00199-f005:**
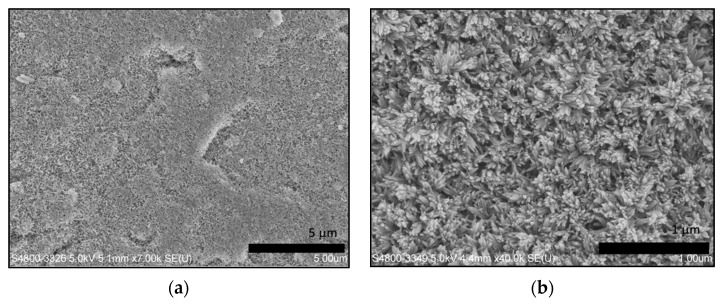
SEM micrograph of remineralized dentin surface in the experimental group with addition of GO: (**a**) with 7.00 k × magnification; (**b**) with 40.0 k × magnification.

**Figure 6 materials-15-00199-f006:**
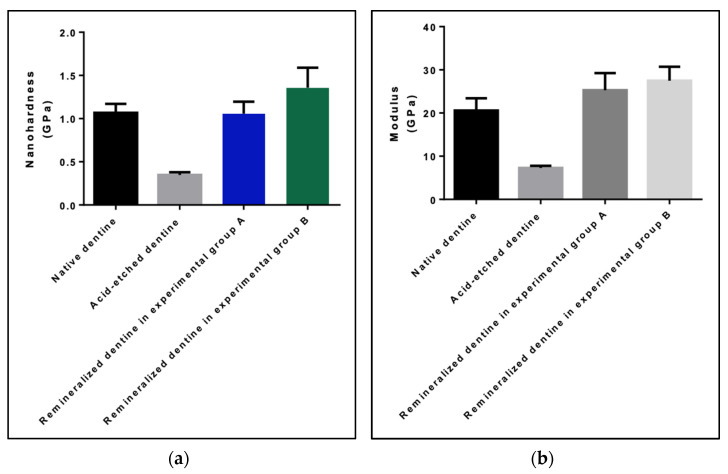
(**a**) Nanohardness and (**b**) modulus of native dentin slices, acid-etched dentin slices, remineralized dentin slices in experimental group A—without addition of GO, and remineralized dentin slices in experimental group B—with addition of GO.

**Table 1 materials-15-00199-t001:** Average concentration (wt.%) of control group (without the aid of electrophoresis) and experimental group (with the aid of electrophoresis) with addition of GO.

Concentration (wt.%)	Control Group—Without the Aid of Electrophoresis	Experimental Group—With the Aid of Electrophoresis and Addition of GO
C	6.198	8.842
Ca	42.524	37.644
P	19.604	18.726
F	4.792	5.223
O	26.883	29.565
Ca/P ratio	1.628	1.533

## Data Availability

The data presented in this study are available on request from the corresponding author.

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
