# Peer review of "Electrophoresis-Aided Biomimetic Mineralization System Using Graphene Oxide for Regeneration of Hydroxyapatite on Dentin"

_materials, 2021, doi:10.3390/ma15010199_

Round 1
Reviewer 1 Report
- There was no continuity or flow of thoughts between paragraph 6 and 7 in introduction.
- Line 145, mineralization cycles were mentioned up to six cycles only. But in the results refer to line 213, remineralization procedures was observed for 8 hrs. Please clarify.
- Line 149, the abbreviation should be FESEM not SEM
- Figure 1 is unclear on the peaks showed in circles. It is better to mark each peaks with wavelength numbers. Label for Y-axis as Transmittance % was also missing.
- Figure 2, missing both labels for X & Y axis. What are the black arrows refer to?
- The results also unclear either the GO with 0.4 or 0.8% GO was giving a good result.
- The results for 2, 4 and 6 hrs remineralization were also not reported.
- Why SEM magnifications used in fig 4a and b was different? Both images were supposed to be in comparison but presented in different magnifications the data may misleading the result interpreted. Need to add scale bar need in each SEM micrograph.
- Similarly to fig 5a & b and figure 6a & b, need to add scale bar need in each SEM micrograph.
- Under 3.2 section, the results were quite ambiguous as the author refer to nanohardness, however the measurement was evaluated using Koop micro hardness tester. Hence need to clarify further. Also need to rewrite on the hardness value up to 3 significant decimal points only or refer to journal’s instruction to authors. Please relook on “+” symbol.
- Line 205, spelling error for surface.
- Line 213, remineralization procedures was observed for 8 hrs or 8 days? In the methodology stated for 8 days, need a clarification.
- Line 277, misspelled of existence.
Author Response
Thank you for the comments
Please find the response attached

Reviewer 2 Report
How does GO infleunce mineralization of dentin, its the premise for the study, but there is no strong evidence provided for this effect, explain with strong evidence of original studies. the aim and hypothesis are not aligned. acceleration or promotion of crystals , it is not clear.you are not treating dentin hypersensitivity in this study do no need for it in the title. Dentin polishing is not explained.
methods is confusing, it should be divided in subheadings to present which tests were performed and with what methods. synthesis of gO is not presented. what process was followed for gO synthesis.
the peaks for GO are usually at 1716 1635 1154, however it si not clear where the peaks are for GO in the FTIR spectra provided, provide a properly marked spectra for identificaiton of peak marks in the x axis. ALso show the complete spectra as provided by the FTIR, this appears to be a section of the complete spectra, y axis is not visible.The typical peaks for identification of GO are usually obsevred in Raman spectroscopy . kindly undertake raman, XRD for confirmation and adequate characterization. The EDS presented does not signify much, where was the gO in the membrane. how much, it does not even show the percentages of elements, as not numbers or y axis is presented. the figure 3 is not contributing or showing anything. SHow percentages of elements and perform line edx to show distribution of GO.
The SEM images are also biased.add original images from the SEM, not sections of images. there is no mention of magnification on the image or labelling on the images to explain what is observed. a looks zoomed and b is zoomed out image showing no dentin structure. and provide multiple images of all the groups. again fig 5 and 6 are different magnification images which could be same images at high and low magnifications. provide original images form the SEM not the modified ones with magni and labelling.
present numbers for mechanical testing in the results as table. the methods for MOE is not mentioned in methods.
For remineralization asessment the author shoudl present Average of calcium, phosphor % and Ca/P wt % in numbers as a table. kindly see previous remin studies.
Kindly provide the suggested and submit for re-evaluation.
thank you
Author Response

(The authors gave the same response as above.)

Reviewer 3 Report
This is a very interesting research that aimed to promote HA crystal
growth using graphene oxide incorporated electrophoresis system. The abstract should be improved and talk more about graphene in the abstract.
There are some comments below to improve all manuscript before be accept.
Introduction section
“However, none of these agents is established as gold-standard 41 for tubule occlusion due to varied results.” Authors should develop this phrase. Why none of these agents is considered a gold-standard….I believe “varied results” is superficial.
The phrases in all introduction are quite short. For example: “Hydroxyapatite (HA) has been known for decades to reduce dentin hypersensitivity. 43 HA-particles have high polarity and they are able to bind to both collagen and HA from dentin. HA-particles occlude dentin tubules by being pressed into the tubule opening.” Please, rewrite some short phrases and develop it better.
“The mineralization will cause fewer tubule opening, thus de- 47 creasing dentin hypersensitivity”. How does it occurs?
Biomimetic 56 mineralization strategies have been proved capable of regenerating HA in vitro. There is no reference.
“The strong hydrophilicity of GO 78 guarantees that it is a good candidate for many applications”. Please, review this phrase and properties of graphene. It is known that graphene sheets present strong hydrophobicity, the main reason for the aggregation of graphene into a composite.
Is the Electrophoresis-aided mineralization system viable for use in a clinical practice? How is the cost? Please, add in discussion section a paragraph about the cost benefit of the system to be use daily in a clinical scenario.
Materials and methods
Dentin slices were placed in a demineralization solution (add reference)
“to create lesions of 70 - 100 mm 99 deep”. How did authors measure it?
Please, add reference for pH-cycling
Author Response

(The authors gave the same response as above.)

Round 2
Reviewer 2 Report
The english language in the abstract is incorrect. "was significantly higher on the group" in the or for the groups.
the aim of the study is not mentioned in the abstract.
the abstract should be an accurate presentation of the study performed and should be a stand alone description which must describe the complete summary of the study. in its current form it is missing a lot of the study aspects. revise. is it dentin or dentine, the author is using both ?
line 73, the english (have been) its incorrect. line 68, too long a sentence. line 82 to 87. the sentence are non specific , do not explain what the author is saying, e.g, electro--- enables ion migration in one directional dimension. (what is the author talking about here, is it in dentin , which ions, its just a vague sentence. next,
Numerous in vitro studies have proved the efficacy of the system (which system), and then u start talking about in vivo study syddenly whithouth expalinig anything about in-vitro studies. "in which an in vivo study by Zhang et al." the focus should be on the mineralization, and the systems safety is being discussed. why ?
88 and 89 the sentence are presentign propertis of graphene, shoudl be GO from the start of para. there are 3 sentences, just introducin GO inthsi para, this is poor writing, the auhtors shoudl tlak about the mineralization effect of GO, the sentence making is poor, use of short sentences is incorrect, and there are no references also. greatly promotes (incorrect english). grahene mat have good biocompatability (incorrect english), line 98 99 100, the sentence has no punctuations, the author is just .
the SEM figure are still poor, as i requested earlier, they should be same magnifications in images for different groups, otherwise how can we evaluate if there is a significant effect, and the magnificantions are also far apart. judging from the images, and magnifications mentioned, different machines are being used in the SEM, however it i think 3 machines are mentioned in the methods. the machines and materials details are nto presented in a standard format, no city country.
with poor images , the paper is not good enough for publication. figure 4 b does not show anything, it shows craked specimen surface. authors were requested to add labels on the SEM figures, whihc is not added. in addition, the distribution of elements in EDS is not shown, whihc was requested. the EDS, figure font is too small. the numbers of % on EDS should b mentioned on the image as provided by the SEM machine. also the SEM micrograph image of area selected for EDS is standard to be included. i requested line EDS for distrobution of elements in the first comments ?
the defineitionsof groups is different in methods and figures, which is confusing. figure 3 group a and b, and in text, line 188-192, concentration table 1, should be presented as groups of the study as defined in text.
The write up is so confusing and without an uniformity. The figures for SEM, must be revised so that comparisons are performed for the effect of elctropho and GO uing same mag of images between groups.
the authors added nizami study in discussion, however this study does not assess mineralization in response to GO , but they asses protection from decalcification using Functionalized GO. therefore th authors must mention the differences and similarities in the studies.
there is not reason for mentioning the authors previous study, they should base future recommendation based on the present study findings. Kindly check references for their purpose.
Author Response
Please see attached our revision report.
Thank you.

Round 3
Reviewer 2 Report
The abstract still misses the quality , it is required for manuscript writing that the abstract should be stand alone and it shud present all vital information in it.unfortunately that is not the case. the first 3 sentences are just the background which is not needed to such an extent. the aim is incomplete it should mention tooth dentin. the methods are not present int he abstract only the groups are mentioned. how was HA identified , is not mentioned.
Kindly use professional writing help for presenting the study as a paper. This is my suggestion. the writ eis still missing mention of references
Author Response
Please find attached our revision report.
Thank you.
